# Neutral Residues: Revisiting Adapters for Model Extension

**Franck SIGNE TALLA** [1]   **Édouard Grave** [1]   **Hervé Jégou** [1]

## Abstract

We address the problem of *extending* a pretrained large language model to a new domain that was not seen during training. Standard techniques, such as finetuning or low-rank adaptation (LoRA) are successful at domain adaptation, but do not formally add capacity to the model. This often leads to a trade-off, between performing well on the new domain *vs.* degrading performance on the original domain. Here, we revisit and improve adapters to extend LLMs from three angles: data, architecture and training procedure, which are advantageously considered jointly. The resulting method, called neutral residues, modifies adapters in a way that leads each new residual block to output near-zeros on the original domain. This solution leads to strong results when adapting a state-of-the-art model originally trained on English to a new language. Neutral residues significantly outperform competing approaches such as finetuning, LoRA or vanilla adapters in terms of the trade-off between learning the new language and not forgetting English.

## 1. Introduction

The dominating strategy for producing foundation models involves training from scratch on a large collection of data, typically trillions of tokens, covering multiple domains. Training such models is extremely costly in terms of computing resources. In view of the explosion of the number of gigantic models produced in the last years (the HuggingFace model repository claims to host about one million models), this training paradigm raises the problem of the economical and ecological sustainability of the model production pipelines. As an example, the cost of training the Llama 3 model is estimated be in the order of hundreds of millions

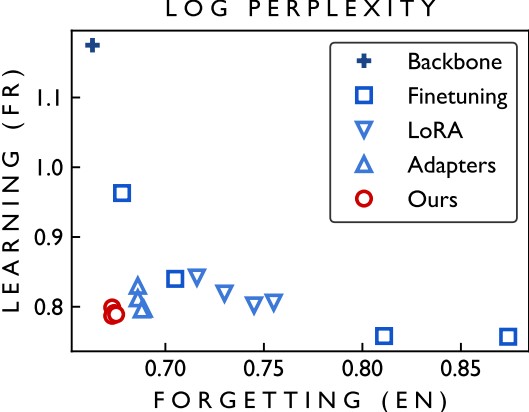

LOG PERPLEXITY

*Figure 1.* **Learning or Forgetting?** Fine-tuning a model reduces the model performance on the original task. Potential trade-offs are governed by the learning rate, with extreme cases being (*top-left*) the original backbone and (*bottom-right*) finetuning on the new data with a large learning rate. LoRA or vanilla adapters mitigate catastrophic forgetting but only to some extent. Our method significantly improves this compromise. We report detailed results in Table 7 of Section 4.

dollars, mostly due to the training on 16,000 H100 GPUs for months,[1] not counting the hidden costs of exploration.

In this paper, we consider the question of extending an existing model, in order to add some new capabilities or knowledge without retraining it from scratch and therefore in a much more resource-efficient manner. The solutions that are successful for domain adaptation, such as finetuning or Low-Rank Adaptation (Hu et al., 2022, LoRA), are not adequate in this context because they do not add any extra capacity. Therefore they are inherently limited in the amount of knowledge that they can incorporate without suffering significant forgetting. This catastrophic forgetting is a well-known problem in the continual learning setting (McCloskey & Cohen, 1989; French, 1999).

In contrast, the adapters introduced by Rebuffi et al. (2017) extend a pretrained model by adding new parameters. When transferring to a new domain or task, only these new weights are trained while those of the original backbone are frozen. This extension strategy, which was initially introduced in

[1]Kyutai, Paris, France. Correspondence to: Franck SIGNE TALLA <franck.signe-talla@kyutai.org>.

*Proceedings of the 42$^{nd}$ International Conference on Machine Learning*, Vancouver, Canada. PMLR 267, 2025. Copyright 2025 by the author(s).

[1]https://ai.meta.com/blog/meta-llama-3/

computer vision with convolutional neural networks, was subsequently adapted to textual transformers by Houlsby et al. (2019). Adapters benefit from the initial pretrained model and hence exhibit competitive transfer performance. They enjoy several additional properties. First, only a subset of weights is trained. Most importantly, they formally add capacity (unlike linear additive adapters like LoRA) and hence do not suffer from the aforementioned limitations. However, adapters still suffer significant forgetting in their current form and are therefore not sufficient for our problem.

In this paper, we build on adapters and, accordingly, increase the capacity of the model by adding feed-forward blocks, while offering a modular design (Pfeiffer et al., 2020). In particular, we consider the use-case where we add a new language to a pretrained model. We measure how the extended model performs on the training criterion, namely perplexity, but also on downstream tasks such as question answering.

We analyze how to optimize the compromise between learning a new language, and not forgetting the initial knowledge of the pretrained model. We study the impact of (1) training data, (2) training strategy, such as losses specifically intended at reducing the forgetting, and (3) architecture. This leads us to identify several important factors for successfully extending a model with adapters:

- **Data:** When learning adapters, a simple way to reduce the forgetting is to keep training with a small fraction of data similar to the original distribution.

- **Architecture:** An *adapter gating* mechanism is a way to ensure that the network distinguishes when it should operate as the original neural network or when the new blocks are desirable to process the input data.

- **Training:** The gating mechanism is advantageously informed in a supervised manner. We proposed two strategies in that respect, both introducing a local loss: The first one is inspired by mixture of experts (MoE), which involves an explicit domain classifier at each block. The second one involves a sparsity loss whose objective is to ensure that the residual connections output near-zero values when the input follows the pretraining distribution.

- **Initialization:** Our analysis concurs with the observation by Houlsby et al. (2019) that near-identical initialization is important when training adapters. We introduce a variant that is even more drastic that the existing 0-block initialization of the output matrix.

Overall, we show that these multiple ingredients are advantageously intertwined to obtain the best trade-off between learning the new distribution well and not forgetting the original capabilities. This leads to our proposed method,

called *neutral residues*. We then apply *neutral residues* to improve the performance of the Gemma LLM (Gemma et al., 2024) on languages such as Danish, Hungarian or Slovak, without degrading the performance on English.

## 2. Related Work

The success of deep learning is often associated to the outstanding performance obtained when training convolutional neural networks on large datasets such as Imagenet (Deng et al., 2009). Yet another key benefit is their amenability to transfer knowledge between tasks with a limited amount of training data (Oquab et al., 2014). This transfer is usually performed with a so-called *finetuning* stage, where the original backbone is trained in a data-efficient manner to solve a distinct target task than the one employed at training time. It often requires to modify the architecture to adapt the output layers to the new tasks. Several self-supervised pretraining methods are noticeably interested in improving ways of pretraining models on proxy tasks (Devlin et al., 2018; Caron et al., 2021), such that networks finetuned from these backbones, or used in a 0-shot manner, perform well on multiple tasks. In most of these settings, the backbone only requires some form of domain adaptation.

Finetuning is however not sufficient when a large body of knowledge must be incorporated to the network. In this section we review a body of literature related to this problem and to our work. This includes solutions that add new knowledge into a pretrained network, as well as methods that incorporate gating mechanisms such as MoE.

**Parameter Efficient Finetuning.** Enhancing language model capabilities using module based training has gained interest in recent years due to the high cost of full finetuning. Those methods are known as parameter efficient finetuning methods (PEFT; Houlsby et al. (2019); Hu et al. (2022); Li & Liang (2021)). Unlike full finetuning, they only require a limited amount of memory. Houlsby et al. (2019) proposed to insert small bottleneck layers, known as Adapters, within each transformer layer, allowing models to be finetuned by training only a small fraction of the parameters: The original model is frozen. There is a large body a literature on adapters, see the overview by Pfeiffer et al. (2023). LoRA (Hu et al., 2022) adds trainable low-rank matrices to transformer layers, while freezing the original weights.

**Continual Learning without Forgetting.** Continual Learning is a widely studied concept (De Lange et al., 2021; Wang et al., 2024) as it allows the addition of new knowledge to LLM after their initial pretraining. It is usually done through full finetuning in order to learn domain specific knowledge (Roziere et al., 2023), to enhance instruction-following abilities (Wang et al., 2023) or to align LLM with

human preferences (Ouyang et al., 2022). One of the biggest challenges of continual learning and the main drawback of doing it through full finetuning is catastrophic forgetting (Kirkpatrick et al., 2017). This is partially alleviated by using the initial training data (Robins, 1995), when available, or as a proxy data from a similar distribution.

Several studies have investigated more advanced methods to address forgetting in continual learning (Biderman et al., 2024; Wu et al., 2024; Li et al., 2024; Zhong et al., 2024; Riemer et al., 2019; Li & Hoiem, 2017). Biderman et al. (2024) show that LoRA helps reducing forgetting but is less efficient at learning than full finetuning, especially for distributions far from the LLM pretraining distribution. Wu et al. (2024) proposed a continual learning method for LLMs, which amounts to adding new intermediate transformer blocks in an interleaved manner. This enables the injection of new knowledge while preserving the initial capabilities.

**Mixture of Experts.** MoEs have a long history in neural networks (Yuksel et al., 2012), for instance with set of classifiers where each classifier was specialized for a different task or domain. Residual architectures (He et al., 2016) and in particular transformers (Vaswani et al., 2017) have renewed the interest in this strategy. In this context, they amount to replacing the standard feed-forward networks (FFN) of transformer blocks by a collection of FFNs, referred to as experts. A gating mechanism selectively activates a subset of relevant experts for a given input. This technique has been widely used to reduce the computational cost of inference in large language models (Jiang et al., 2024; Xue et al., 2024) with a focus on the design of an expert-balancing loss to promote equitable token distribution across experts (Shazeer et al., 2017; Pfeiffer et al., 2023).

Recent works have explored utilizing MoE's gating mechanism to mitigate catastrophic forgetting during the continual learning phase of language models. LoRAMoE (Dou et al., 2024) employs LoRAs as experts, integrating them with a gating mechanism to improve performance on specific downstream tasks through Supervised finetuning (SFT). LoRAMoE mitigates catastrophic forgetting in the case of SFT, but to our knowledge its effectiveness has not been studied in a broader context. In contrast, two recent parallel concurrent works are closer to our proposal: Zhong et al. (2024) augment Mixture-of-Experts LLMs with new modalities by addition of new experts to a pretrained mixture of experts. Similarly, Li et al. (2024) add an MoE layer in parallel to the FFN of the transformer block to learn multilingual capabilities and mitigate catastrophic forgetting during Continual Learning. Our solution shares some similarities with this work by employing mixed data training. Yet Li et al. (2024) do not integrate an explicit sparsity criterion, and need to weight differently the blocks of the pretrained model with those of the gated adapters. They also consider multiple

elements in the mixture and apply a softmax for the rooting mechanism. Since at least one expert is activated in their mixture, this selection mechanism also prevents the capacity of the model to produce 0-valued outputs. Our solution of neutral residues with local loss is more effective than simply gating adapters, as shown in our experimental Section 4.

## 3. Neutral Residues with Gated Adapters

The first adapters were initially introduced by Rebuffi et al. (2017), in the context of convolutional residual networks. Their motivation was to adapt a backbone to multiple tasks by finetuning on a limited amount of data, rather than adding a large body of knowledge. Therefore they considered a limited number of trainable parameters, typically less than 10% additional weights compared to the original backbone. In the context of language models, Houlsby et al. (2019) add residual feed-forward block sequentially after each multi-head attention and the original FFN. In our case and as shown in Figure 2, we add more capacity (typically 20%) and adopt parallel adapters as we do not observe a noticeable difference compared to serial adapters. This concurs with observations by Touvron et al. (2022) that blocks can be parallelized pairwise if vision transformers are large enough.

**Mixed distribution training.** As discussed in the related work, a solution to limit catastrophic forgetting is to keep training with a small amount of the original data, if such data is available. This strategy is typically employed with finetuning but is effective with all baselines. Li et al. (2024) report that catastrophic forgetting almost disappears when the volume of data in original language is five times greater than in the new one. Yet they also point out that mixed data training can slow down the learning of the new language.

Unfortunately, in many cases, the original training data is not available when extending a model. Thus, in the following, we do not assume that we have access to the original data distribution, but only to data that is somewhat related. For example, when using a model that was pretrained mostly on English Common Crawl, we use data from English Wikipedia to capture the original distribution. We also consider the more realistic case of using a public model, for which we do not have details about the pretraining data distribution, such as the Gemma model (Gemma et al., 2024). We study the impact of the proportion $p$ of data related to the original domain. As a particuler case, we analyze the case $p = 0$, where no data related to the pretraining is available.

**Local loss & adapter gating.** One way to prevent forgetting the pretraining distribution is to constrain the output of the model so that it does not change on the original data. This can be achieved by forcing the outputs of the adapters

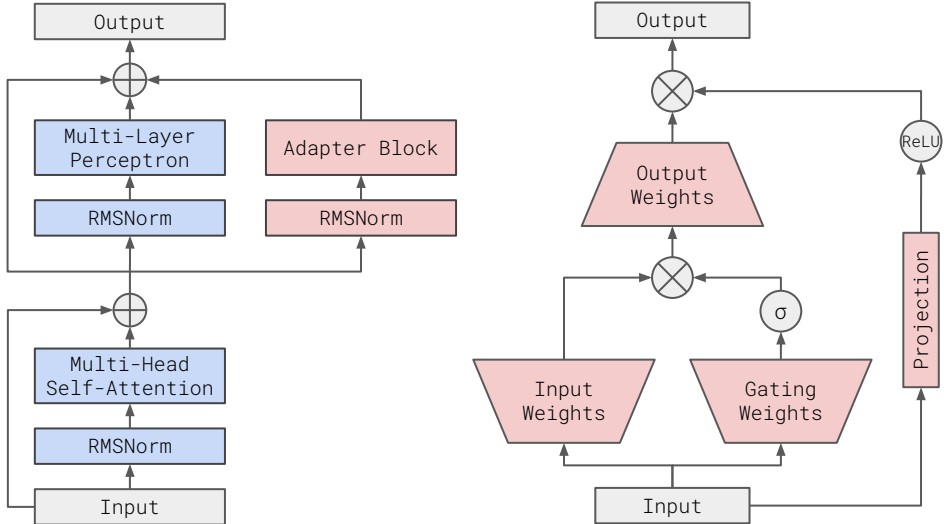

*Figure 2.* **Architectural design** adopted for neutral residues: (*left*) we add the adapter block in parallel to the FFN ; (*right*) our gated adapter block includes a *block gate* with a ReLU non-linearity (see Section 4.4 for a comparison with a sigmoid-based gating). At training time, the weights of the blue blocks are frozen, while the trainable parameters are represented by red blocks. The output of the adapter is trained with a local loss such that the output is sparse if the input follows the pretraining distribution. We note that the $\sigma$ activation function of the adapter block is the same as the one used in the backbone.

to be equal to zero on the original data, and only non-zero on data from the new domain. To this end, we propose to regularize the output of the adapter modules with an $\ell_1$-norm on the data related to the original distribution. We argue that regularizing the output of adapters is easier than continual learning on the original distribution. Hence, using an imperfect dataset representing the original distribution should have a smaller impact on our method than finetuning.

During early experiments, we observed that regularizing the output of adapters helped mitigate forgetting, but it also prevented learning the new domain. As most recent transformer LLMs, the models used in our experiments use gated linear units (GLU) in feed-forward blocks (Dauphin et al., 2017; Shazeer, 2020). The output of the block is obtained with $\text{FFN}(x) = \mathbf{W}_o \left( \sigma \left( \mathbf{W}_g x \right) \odot \mathbf{W}_i x \right)$, where $\odot$ is the pointwise multiplication and $\sigma$ is a non-linearity such as SiLU (Elfwing et al., 2018). We observed that the singular values of the gating weights $\mathbf{W}_g$ of adaptors are significantly more skewed than in regular transformer block from the pre-trained backbone. Figure 3 shows this behavior, which is detrimental to the learning process. Our interpretation is that all the projections instantiated by this operator tend to be colinear, and therefore to behave similarly when combined with the non-linearity, because they implicitly operate as classifier between the old and new distributions.

To address this problem, we add a *block gate* over the full adapter, in a similar fashion as the gating mechanism of MoE transformers (Shazeer et al., 2017), and their application to model extension (Zhong et al., 2024; Li et al., 2024).

In our case, we add a gate only on the additional adapter block, and not on the original block. We optimize the loss $\ell_{\text{train}} = \ell_{\text{LM}} + \alpha \ell_{\text{local}}$, where $\ell_{\text{LM}}$ is the language modeling objective and $\ell_{\text{local}}$ is a loss to train the gating to discriminate the original *vs.* the new distribution. We consider two gating activations and corresponding local losses:

- *ReLU activation with sparsity constraint.* This strategy, which is the one that we refer to as *"neutral residues"* unless specified otherwise, does not make an explicit classification. Instead, we let the ReLU activation adapts itself the strength of its response for the block. We compute the average of the $\ell_1$-norm on the output of the adapters as the local loss. We normalize the loss by the dimension of the model. This loss is only used when training on the original distribution.

- *Sigmoid activation with cross entropy.* This solution is inspired by MoE. In this case, the block gate is associated with the sigmoid activation. It is thus trained as a local binary classifier whose objective is to distinguish between the old and the new data distributions, using the cross-entropy loss averaged over the adapters. In contrast to the previous strategy, this loss is used on both the original and the new distributions.

In Figure 3, observe that the gating operator has a significant effect on the singular values of the gating matrix: the distribution of the singular values is less skewed than the original backbone. Moreover, the gating mechanism based on ReLU

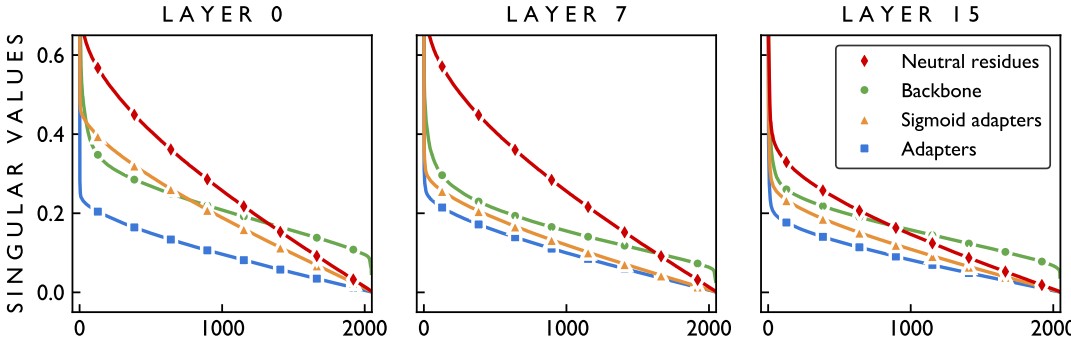

*Figure 3.* **Spectral analysis** of the weight gating matrix of GLU blocks, normalized by the largest singular value.

and $\ell_1$-norm is more effective than the one based on sigmoid and the cross-entropy loss. As we will see in Section 4, it is also the method that offers the best compromise between not forgetting and learning.

**Low-variance initialization.** One key ingredient pointed out by Houlsby et al. (2019) is that it is best to start training adapters such that the initial network is near-identical to the pretrained one, thereby not modifying the initial function. This is typically done by initializing the output adapter matrix to $0s$, as also advocated by Zhang et al. (2019). We are even more drastic in that respect: in addition to setting the output matrix with $0s$, we depart from the usual He's initialization for both the input and gating matrix and initialize them with a much lower variance, such that the model remains longer close to the original one. More specifically, we reduce the variance employed in He's initialization: instead of initializing weights with variance $2/d$, where $d$ in the input dimension of the matrix, we use a variance of $1/(d \cdot L)$, where $L$ is the number of transformer layers.

**FFN *vs*. multi-head attention (MHA).** In our early experiments carried out to add a new language, it is more effective to add extra weights in the form of FFN than in MHA blocks, for a given number of additional parameters. This is consistent with how extra capacity is added in MoE. Similarly Li et al. (2024) present an ablation that shows that adding FFN blocks is better than adding attention blocks. Note, this is in contrast with what is observed by Hu et al. (2022) with the LoRA method, for which the best choice to adapt to different downstream tasks with few parameters is to update the MHA matrix, in particular they choose to update the keys and values matrices. In our preliminary experiments, we re-evaluate LoRA in our context of knowledge addition, and observe that LoRA is more effective when updating the FFN blocks than the MHA blocks. We hypothesize that this apparent discrepancy is due to the fact that task adaptation is different from the task of extending a pretrained network on a large body of knowledge.

## 4. Experiments

In this section, we report experimental results to validate the performance of neutral residues to adapt a neural network to a new domain. We consider the case of adding or improving the multilingual ability of a large language model. More precisely, we start from an English-only model (or a model that has only seen a small amount of non-English data) and finetune it on another target language. Section F covers the case of multiple languages for a multilingual model.

### 4.1. Datasets and evaluation protocols

**Training Datasets.** For the multi-lingual finetuning datasets, we use data extracted from CommonCrawl, and process it with the following steps. First, the text content is extracted from HTML using the `resiliparse` package.[2] Then, we perform language identification with `fastText`[3] to keep data in the target language only. Next, we perform deduplication at the paragraph level, by computing a hash of each paragraph. Finally, we filter document using a `fastText` linear classifier, which was trained to discriminate random pages from CommonCrawl *vs*. high-quality documents such as Wikipedia articles, textbooks, news articles or scientific articles.

For the English domain, we restricted ourselves to text from Wikipedia. The motivation for this is to use a finetuning dataset that is related, but different, from the training dataset of the backbone model. Indeed, as discussed earlier, in many cases one does not have access to the original dataset that was used to train the backbone model, and we want to demonstrate that our method is robust even in the case where exact data from the original distribution is not available.

**Evaluation benchmarks.** We evaluate the performance of finetuning with two steps. First, we evaluate the perplexity

---

[2]https://resiliparse.chatnoir.eu
[3]https://fasttext.cc

*Table 1.* **Mixture of distributions:** impact of the rate $p$ of data that roughly approximate the pretraining distribution when learning a vanilla adapter. 10% of initial data distribution is a good compromise between learning (FR) and not forgetting (EN). We report perplexity (lower is better).

| English rate $p$ | EN | FR |
|---|---|---|
| 0.00 | 0.720 | 0.810 |
| 0.01 | 0.707 | 0.810 |
| 0.10 | 0.687 | 0.812 |
| 0.50 | 0.683 | 0.828 |
| Pretrained model | 0.663 | 1.175 |

*Table 2.* **Training with new data only or mixed data.** Starting from an English pretrained model, we measure the tradeoff between learning and forgetting if we only post-train with target data or if we keep $p = 10\%$ of original data. See Table 1 for results of the pretrained model. We report the perplexity on the validation sets at the end of training.

| Method | $p = 0$ | | $p = 0.1$ | |
|---|---|---|---|---|
| | EN | FR | EN | FR |
| Finetuning | 0.874 | 0.755 | 0.811 | 0.758 |
| LoRA | 0.770 | 0.814 | 0.730 | 0.818 |
| Vanilla adapters | 0.720 | 0.810 | 0.687 | 0.812 |
| Neutral residues | 0.684 | 0.790 | 0.668 | 0.793 |

of the model on held-out sets, in English and in the target language, to measure forgetting and learning. For English, we use text from a domain that does not correspond to the finetuning data: the PubMed subset from ThePile (Gao et al., 2020). The goal here is to evaluate how robust our method is to forgetting, especially in the case where we *do not have access* to the same training distribution as the original model. For target languages, we consider text from the same distribution as the finetuning dataset.

Second, we consider standard academic benchmarks used to evaluate large language models, such as question answering or Cloze-style problems. We use the following datasets: ARC challenge (Clark et al., 2018), HellaSwag (Zellers et al., 2019), MMLU (Hendrycks et al., 2020), Common-Sense QA (Talmor et al., 2018) and Belebele (Bandarkar et al., 2023). For target languages, we use the translated datasets from Dac Lai et al. (2023) and Sakai et al. (2024), when they exist. For all datasets, we score each answer independently as the continuation of the question, and predict the most probable one. We use normalization, following the recommendation from Gu et al. (2024). We perform 5-shot evaluation for ARC challenge, CSQA and MMLU, and 0-shot evaluation for HellaSwag and Belebele.

**Backbone models.** We consider two models that we use as backbone. The first, called **EN-LM-1B**, was trained internally on 2.4T tokens of English data only, and has 1B parameters. The pretraining data consists mostly of web pages filtered from Common Crawl, as described previously. It also includes a small amount of data from curated sources such as Wikimedia, Stack Exchange and scientific articles. The second model is **Gemma-2B** (Gemma et al., 2024) which was pretrained on a small amount of multi-lingual data. We do not have access to Gemma's training data, making it a realistic test case for our approach. Appendix B gives more details about these models. We use **EN-LM-1B** for all the experiments, except for the main results that are reported in Table 3. We chose to use **EN-LM-1B** for our exploratory experiments and ablations as it is smaller, hence

leading to faster experiments.

**Baseline extension strategies.** In our experiments, we use the following methods as baselines:

- *Finetuning:* we train all the weights from the backbone. We re-initialize the optimizer state.

- *LoRA:* unlike Hu et al. (2022), we add all the additional parameters in the FFN layers, as in our preliminary experiments this choice was significantly better for model extension.

- *Adapters:* we use the vanilla adapters as Houlsby et al. (2019). We use parallel instead of serial blocks to have a more directly comparable baseline. We use the standard adapters' initialization, *ie.* zeros for the output matrix and He's initialization for the other weights.

**Hyperparameters.** Except mentioned otherwise, LoRA, adapters, and *neutral residues* use 20% of extra learnable weights. For each method, we selected the learning rate that leads to the best trade-off between learning and forgetting: $5 \cdot 10^{-5}$ for finetuning and LoRA, $2 \cdot 10^{-4}$ for adapters and *neutral residues*. The hyperparameter $\alpha$ governing the strength of the sparsity loss is set by default to 0.01 for *neutral residues*. When training on the new data, we train during 100,000 steps with a batch size of 64 sequences of length 4,096 for both **EN-LM-1B** and **Gemma-2B**. We provide other training hyperparameters in Section B.

### 4.2. Preliminary Analysis

**Training with mixed data distribution** Table 1 analyzes the impact of the ratio $p$ of data approximating the pretraining distribution used to extend the model with vanilla adapters. We note that $p = 0.1$ offers a good trade-off: the forgetting is not significantly higher than with $p = 0.5$, while the model is almost as good in French as when we train with French only ($p = 0$). Thus, when training with mixed data, we set $p = 0.1$ in all subsequent experiments.

*Table 3.* **Main results:** comparison of neutral residues (*ours*) versus four baselines for the Gemma-2B model. All methods except finetuning use 20% of trainable parameters compared with the backbone models. We use a proportion $p = 0.1$ of English data for the training. We report the average performance across all tasks both on English and target languages. These metrics reflect the overall performance with respect to not forgetting and learning, respectively. In **bold** we report the best method, in underline the second best.

| Target | Method | Forgetting: English | | | | | Learning: Target | | | | | Task avg. | |
| | | BeleBele (0) | HellaS (0) | Arc C (5) | CSQA (5) | MMLU (5) | BeleBele (0) | HellaS (0) | Arc C (5) | CSQA (5) | MMLU (5) | Forgetting (English) | Learning (Target) |
|---|---|---|---|---|---|---|---|---|---|---|---|---|---|
| French | Backbone | 46.3 | **69.7** | 47.0 | 63.0 | **40.2** | 44.3 | 50.3 | 40.9 | 53.3 | 33.9 | 53.2 | 44.6 |
| | Finetuning | 45.7 | 62.7 | 45.9 | 57.3 | 37.2 | 44.9 | **60.8** | **43.9** | 63.8 | 35.9 | 49.8 | **49.9** |
| | LoRA | 46.1 | 65.7 | 45.2 | 60.5 | 38.4 | 46.3 | 57.1 | 41.7 | 44.2 | 35.1 | 51.2 | 44.9 |
| | Adapters | 48.1 | 67.8 | **47.5** | 61.4 | 39.3 | 44.6 | 57.4 | 35.2 | 56.9 | 36.0 | 52.8 | 46.0 |
| | *Ours* | **48.2** | 68.5 | 47.3 | **63.3** | 39.2 | **46.8** | 57.9 | 39.7 | 59.8 | **36.7** | **53.3** | 48.2 |
| Danish | Backbone | 46.3 | **69.7** | **47.0** | 63.0 | **40.2** | 41.6 | 43.1 | 35.6 | N/A | 32.3 | **53.2** | 38.1 |
| | Finetuning | 46.3 | 62.3 | 37.9 | 55.1 | 34.5 | 41.3 | **57.9** | **39.2** | N/A | 32.8 | 47.2 | 42.8 |
| | LoRA | 45.2 | 64.7 | 44.7 | 52.4 | 35.3 | 42.2 | 55.3 | 37.4 | N/A | 32.9 | 48.5 | 41.9 |
| | Adapters | **46.9** | 66.8 | 45.9 | 57.1 | 37.9 | 42.3 | 53.7 | 39.0 | N/A | 34.2 | 50.9 | 42.3 |
| | *Ours* | 45.4 | 69.4 | 46.6 | 60.4 | 38.1 | **42.4** | 55.8 | 38.7 | N/A | **34.7** | 52.0 | **42.9** |
| Hungarian | Backbone | 46.3 | **69.7** | 47.0 | 63.0 | **40.2** | 34.3 | 36.0 | 31.1 | N/A | 31.0 | **53.2** | 33.1 |
| | Finetuning | 44.6 | 60.3 | 40.8 | 48.5 | 35.2 | 37.1 | **48.1** | 36.1 | N/A | 32.5 | 45.9 | 38.5 |
| | LoRA | 45.0 | 63.6 | 43.3 | 54.6 | 36.3 | 39.0 | 44.9 | 35.4 | N/A | 32.7 | 48.6 | 38.0 |
| | Adapters | **48.0** | 66.5 | 48.1 | 59.1 | 38.1 | **39.6** | 44.8 | 36.6 | N/A | 32.7 | 52.0 | 38.4 |
| | *Ours* | 46.1 | 69.2 | **48.5** | 60.6 | 38.5 | 38.2 | 46.1 | **38.1** | N/A | 32.8 | 52.6 | **38.8** |
| Slovak | Backbone | 46.3 | **69.7** | **47.0** | **63.0** | **40.2** | 38.1 | 38.8 | 32.8 | N/A | 31.3 | **53.2** | 35.2 |
| | Finetuning | 43.2 | 60.0 | 40.0 | 54.3 | 34.3 | 41.3 | **47.8** | **35.2** | N/A | 32.4 | 46.4 | **39.2** |
| | LoRA | 44.8 | 63.8 | 41.7 | 54.9 | 36.0 | 41.4 | 46.7 | 32.9 | N/A | 32.3 | 48.2 | 38.4 |
| | Adapters | 45.2 | 62.5 | 38.0 | 55.9 | 37.4 | **42.3** | 40.9 | 34.9 | N/A | **32.7** | 47.8 | 37.7 |
| | *Ours* | **47.9** | 68.0 | 42.8 | 60.6 | 38.1 | 40.8 | 46.6 | 34.7 | N/A | 32.2 | 51.5 | 38.6 |

*Table 4.* **Ablation: Gating and local loss.** We report the average across tasks; see Table 16 for details.

| | Gate | Gate loss | | Perplexity ↓ | | Tasks ↑ | |
| | | $\ell_1$ | CE | EN | FR | EN | FR |
|---|---|---|---|---|---|---|---|
| (1) | ∅ | ✓ | | 0.687 | 0.801 | 45.3 | 42.4 |
| (2) | Sigmoid | | ✓ | 0.677 | 0.800 | 45.2 | 42.6 |
| (3) | | ✓ | ✓ | 0.676 | 0.800 | 45.3 | 41.9 |
| (4) | ReLU | | | 0.673 | 0.791 | 46.4 | 42.8 |
| (5) | | ✓ | | 0.674 | 0.791 | 47.1 | 43.6 |
| (6) | Adapter baseline | | | 0.686 | 0.812 | 45.1 | 41.3 |
| (7) | Backbone baseline | | | 0.663 | 1.175 | 47.0 | 32.9 |

Table 2 gives the trade-off between learning and not forgetting for all baselines and our method, when training either without or with mixed training distributions. As one can see, the mixed training significantly reduces the forgetting for all the methods without hindering too much the learning. We report more extensive results in Table 10 of the appendix, including on downstream tasks, when evaluating the impact of including some data related to the original distribution.

### 4.3. Main results

We report our main results in Table 3, where we extend the capabilities of the Gemma 2B model on four languages: Danish, French, Hungarian and Slovak. First, we note that our method, *neutral residues*, outperforms the LoRA and adapters baselines on all four languages. In average, it is always among the two best methods: it is only second to the backbone on forgetting and to finetuning on learning. However, *neutral residues* offers a significantly better trade-off than both these methods. Second, we observe that adapters tend to perform better than LoRA on the task of extending Gemma 2B to a target language. This is a confirmation that for tasks requiring significant new knowledge, it is helpful to explicitly add capacity to the base model in the form of additional parameters. This observation is confirmed when varying the number of additional parameters: we observe improvements when increasing the extra capacity from 5% to 50% (we report these results in Table 9 of the Appendix).

### 4.4. Ablations

**Ablation of the gating and local loss.** In Table 4, we compare adapters using different gating mechanisms, and

*Table 5.* **Ablation: initialization.** We report the average across tasks; see Table 17 for details.

| Gate | Our init. | Perplexity ↓ | | Tasks avg. ↑ | |
|---|---|---|---|---|---|
| | | EN | FR | EN | FR |
| Adapters w/ $\ell_1$-norm | | 0.687 | 0.812 | 45.5 | 41.6 |
| | ✓ | 0.687 | 0.801 | 45.3 | 42.4 |
| Adapters w/ sigmoid | | 0.668 | 0.810 | 45.6 | 40.8 |
| | ✓ | 0.677 | 0.800 | 45.2 | 42.6 |
| Neutral Residues | | 0.673 | 0.818 | 46.5 | 41.6 |
| | ✓ | 0.674 | 0.791 | 47.1 | 43.6 |

*Table 6.* **Ablation:** $\alpha$ governs the strength of the $\ell_1$ local loss. See Table 18 for detailed results.

| | Perplexity ↓ | | Tasks avg. ↑ | |
|---|---|---|---|---|
| $\alpha$ | EN | FR | EN | FR |
| 0 | 0.673 | 0.791 | 46.4 | 42.8 |
| 0.01 | 0.674 | 0.791 | 47.1 | 43.6 |
| 0.1 | 0.672 | 0.791 | 46.1 | 42.9 |
| 1 | 0.674 | 0.791 | 46.7 | 42.3 |
| 10 | 0.672 | 0.791 | 46.6 | 43.2 |
| 100 | 0.667 | 0.791 | 46.3 | 43.0 |

*Table 7.* **Trade-offs with different learning rates.** This hyperparameter allows us to favor learning *vs.* forgetting or vice-versa. These results are the ones reported in Figure 1. See Table 19 for detailed results.

| Method | LR | Perplexity ↓ | | Tasks avg. ↑ | |
|---|---|---|---|---|---|
| | | EN | FR | EN | FR |
| Backbone | 0 | 0.663 | 1.175 | 47.0 | 32.9 |
| Fine-tuning | $2 \cdot 10^{-5}$ | 0.678 | 0.963 | 45.2 | 36.3 |
| | $5 \cdot 10^{-5}$ | 0.705 | 0.840 | 44.4 | 41.1 |
| | $2 \cdot 10^{-4}$ | 0.811 | 0.758 | 38.9 | 43.6 |
| | $5 \cdot 10^{-4}$ | 0.874 | 0.757 | 36.0 | 42.5 |
| LoRA | $2 \cdot 10^{-5}$ | 0.716 | 0.842 | 44.1 | 40.5 |
| | $5 \cdot 10^{-5}$ | 0.730 | 0.819 | 43.7 | 41.2 |
| | $2 \cdot 10^{-4}$ | 0.745 | 0.802 | 41.9 | 41.7 |
| | $5 \cdot 10^{-4}$ | 0.755 | 0.806 | 41.6 | 42.6 |
| Vanilla adapters | $2 \cdot 10^{-5}$ | 0.686 | 0.830 | 45.1 | 40.8 |
| | $5 \cdot 10^{-5}$ | 0.686 | 0.812 | 45.1 | 41.3 |
| | $2 \cdot 10^{-4}$ | 0.689 | 0.797 | 46.0 | 42.4 |
| | $5 \cdot 10^{-4}$ | 0.688 | 0.796 | 45.4 | 43.5 |
| Neutral residues | $2 \cdot 10^{-5}$ | 0.673 | 0.799 | 46.4 | 42.9 |
| | $5 \cdot 10^{-5}$ | 0.674 | 0.791 | 47.1 | 43.6 |
| | $2 \cdot 10^{-4}$ | 0.673 | 0.787 | 46.4 | 42.5 |
| | $5 \cdot 10^{-4}$ | 0.675 | 0.789 | 46.6 | 43.0 |

different loss functions to learn the gating. First, comparing line (5) to line (1), we observe that regularizing the output of the adapters alone, without adding a block gating mechanism does not work well. We hypothesize that this is mainly due to the weights of the adapter to become co-linear, as discussed in spectral analysis of Section 3. In particular, this prevent the adapter to learn the new domain well. Next, we observe that the combination of the ReLU activation with $\ell_1$-norm outperforms the sigmoid activation with cross entropy loss (line (5) *vs.* (2)). Finally, comparing lines (4) and (5), we note that adding supervision with the $\ell_1$ regularization helps learning adapters, even though the ReLU gating alone already leads to strong results.

**Initialization ablation.** In Table 5, we report the performance of various adapters, with and without the low-variance initialization described in Section 3. Overall, we note that this initialization improves the results of gated adapters. More specifically, compared to He's initialization, it leads to better learning of the new target domain.

**Learning rate ablation.** Table 7 reports the trade-offs achievable by different techniques when varying the learning rate. The results of our method are stable across a large range of learning rates (one order of magnitude from $5 \cdot 10^{-5}$ and $5 \cdot 10^{-4}$), in sharp contrast with finetuning and LoRA, which are highly sensitive to this parameter.

**Coefficient $\alpha$ ablation.** Table 6 shows that even without the sparsity constraint, ReLU gating enables a good balance between learning and forgetting. Introducing this constraint further improves the trade-off, with low values ($\alpha = 0.01$) yielding the best results. However, a good trade-off is maintained across a wide range of strengths.

## 5. Conclusion

This paper has explored the feasibility and effectiveness of extending existing foundation models to incorporate new capabilities or knowledge without the need for resource-intensive retraining from scratch. By building upon the concept of adapters, which add new parameters to a pre-trained model, we have demonstrated that it is possible to extend a model without compromising its original knowledge, thereby offering a more sustainable approach than retraining from scratch. Our study focused on the use-case of adding a new language to a pretrained model and evaluated the performance of the extended model on both training criteria and downstream tasks. Our findings highlight several critical factors that contribute to the successful extension of a model while mitigating the issue of catastrophic forgetting. These factors include the strategic combination of mixed training data, an adapter gating mechanism coupled with a local loss, and the importance of near-identical initialization. Our method, called *neutral residues*, obtains state-of-the-art results on extending multilingual capabilities of Gemma-2B.

## Impact Statement

The goal of our work is to advance the efficiency of large langage models (LLMs) training, by improving methods to extend an existing model to a domain requiring a large amount of new knowledge. This has the potential to improve the sustainability and the accessibility of these systems, across various applications and domains. In particular, this has potential environmental implications, as it could reduce the computational resources needed to develop a model for a new domain. Instead of training the model from scratch, one could start from a strong existing base model and use our method to extend it to the target domain. Our method can also improve the accessibility of these systems, by allowing organizations and researchers to personalize existing models to their need without requiring a large amount of computational resources.

In general, the widespread application of LLMs has many other potential social consequences, including risks. It requires careful considerations of ethical issues around fairness, transparency and potential misuse. As our research do not directly address any of these, we do not feel any of these risks must be specifically highlighted here. We thus refer the reader to previous work on the matter, such as Bommasani et al. (2021) or Bender et al. (2021).

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

## A. Architecture details and Hyper-parameter settings

We consider a vanilla transformer **EN-LM-1B**, which was trained internally on 2.4T tokens of English data only, and has 1B parameters. We also considered the Gemma 2B model (Gemma et al., 2024) and Gemma2 2B model (Team et al., 2024), for which we do not know the distribution. For these models, we refer to their paper for training hyper-parameters.

*Table 8.* **Hyper-parameters details**

|  | **EN-LM-1B** | **Gemma 2B** | **Gemma2 2B** |
|---|---|---|---|
| Number of layers $L$ | 16 | 18 | 18 |
| Working dimensionality | 2048 | 2048 | 2048 |
| Number of heads | 128 | 256 | 256 |
| Dimension FFN latent space | 5632 | 16384 | 16384 |
| Activation type | Gated SILU | Gated GELU | Gated GELU |
| Normalization | RMS pre-norm | RMS pre-norm | RMS pre & post norm |
| Group query |  | ✓ | ✓ |
| Pretraining context length | 4096 | 8192 | 8192 |
| **Hyper-parameters for training the extended model** | | | |
| Batch size | 64 | 64 | 64 |
| Context length | 4096 | 4096 | 4096 |
| #Training steps | 100,000 | 100,000 | 100,000 |
| AdamW: $\beta_1$ | 0.9 | 0.9 | 0.9 |
| AdamW: $\beta_2$ | 0.95 | 0.95 | 0.95 |

## B. Impact of the number of parameters

We vary the extra capacity allocated to new learnable weights in LoRA, adapters and *neutral residues (ours)*. The steady performance improvement in the new language highlights the importance of expanding the base model's capacity for tasks requiring substantial new knowledge. Interestingly, the impact of extra weights on forgetting is almost neutral. We trained **EN-LM-1B** with a learning rate of $2 \cdot 10^{-4}$ for finetuning and adapters, and $5 \cdot 10^{-5}$ for LoRA and *ours* with other hyperparameters detailed in Table 8. For each extra weight size, we reported in **bold** the best method.

*Table 9.* **Impact of the number of parameters.**

| Extra Weights | Method | Forgetting: English | | | | | Learning: French | | | | | Task avg. | |
|---|---|---|---|---|---|---|---|---|---|---|---|---|---|
| | | *BeleBele (0)* | *HellaS (0)* | *Arc C (5)* | *CSQA (5)* | *MMLU (5)* | *BeleBele (0)* | *HellaS (0)* | *Arc C (5)* | *CSQA (5)* | *MMLU (5)* | *English* | *French* |
| | Backbone | 43.0 | 60.3 | 40.5 | 56.5 | 34.5 | 34.6 | 32.8 | 26.7 | 42.3 | 28.0 | 47.0 | 32.9 |
| | Finetuning | 37.9 | 45.5 | 34.8 | 45.5 | 30.9 | 42.2 | 48.6 | 35.3 | 60.3 | 31.5 | 38.9 | 43.6 |
| +5% | LoRA | 42.0 | 54.1 | 36.1 | 50.1 | 32.8 | 41.7 | 42.3 | 32.4 | 50.0 | 30.4 | 43.0 | 39.4 |
| | Adapters | 42.2 | 57.9 | **40.4** | 55.1 | 33.7 | **44.0** | **44.1** | **34.2** | 49.5 | **30.5** | 45.9 | 40.5 |
| | *ours* | **42.9** | **59.3** | 40.1 | **55.8** | **33.8** | 42.2 | **44.1** | 33.6 | **52.1** | 30.3 | **46.4** | **40.5** |
| +10% | LoRA | 41.7 | 53.6 | 38.1 | 51.1 | 32.8 | 42.4 | 44.2 | 32.2 | 51.8 | 30.3 | 43.5 | 40.2 |
| | Adapters | 42.9 | 57.5 | 38.7 | 53.3 | 33.6 | 43.2 | 46.4 | 33.5 | 53.4 | 30.7 | 45.2 | 41.5 |
| | *ours* | **44.4** | **58.9** | **42.2** | **56.8** | **34.2** | **44.3** | **46.5** | **34.1** | **56.5** | **31.0** | **47.3** | **42.5** |
| +20% | LoRA | 42.0 | 53.1 | 38.1 | 52.3 | 32.8 | 42.4 | 46.2 | 32.7 | 54.2 | 30.7 | 43.7 | 41.2 |
| | Adapters | 43.1 | 57.7 | 40.3 | 54.1 | **34.5** | 44.3 | **48.8** | 34.8 | 52.7 | 31.2 | 46.0 | 42.4 |
| | *ours* | **43.9** | **59.3** | **41.6** | **56.8** | 34.1 | **45.1** | **48.8** | **35.7** | **56.9** | **31.6** | **47.1** | **43.6** |
| +50% | LoRA | 41.8 | 52.0 | 36.1 | 51.0 | 32.7 | 44.3 | 47.9 | 35.6 | 56.2 | 30.8 | 42.7 | 43.0 |
| | Adapters | **43.3** | 57.6 | 39.4 | 53.3 | 33.5 | **45.8** | 50.4 | **36.5** | 58.0 | 31.9 | 45.4 | **44.5** |
| | *ours* | 43.1 | **59.2** | 40.3 | **56.3** | **34.2** | 43.8 | **50.9** | 35.2 | **59.3** | **32.1** | **46.6** | 44.3 |

## C. Impact of mixed data distribution.

Table 10 shows the importance of training with data approximating the pretraining distribution of the model. The mixed training significantly reduces the forgetting for all the methods without hindering too much the learning. We trained **EN-LM-1B** with a learning rate of $2 \cdot 10^{-4}$ for finetuning and adapters, and $5 \cdot 10^{-5}$ for LoRA and *ours* with other hyperparameters detailed in Table 8. In **bold** we report the best method, in underline the second best.

Table 10. **Impact of mixed data distribution.**

| Ratio | Method | Forgetting: English | | | | | Learning: French | | | | | Task avg. | |
| | | BeleBele (0) | HellaS (0) | Arc C (5) | CSQA (5) | MMLU (5) | BeleBele (0) | HellaS (0) | Arc C (5) | CSQA (5) | MMLU (5) | English | French |
|---|---|---|---|---|---|---|---|---|---|---|---|---|---|
| $p = 0.1$ | Backbone | 43.0 | **60.3** | 40.5 | 56.5 | **34.5** | 34.6 | 32.8 | 26.7 | 42.3 | 28.0 | 47.0 | 32.9 |
| | Finetuning | 37.9 | 45.5 | 34.8 | 45.5 | 30.9 | 42.2 | 48.6 | 35.3 | **60.3** | 31.5 | 38.9 | 43.6 |
| | LoRA | 42.0 | 53.1 | 38.1 | 52.3 | 32.8 | 42.4 | 46.2 | 32.7 | 54.2 | 30.7 | 43.7 | 41.2 |
| | Adapters | 43.1 | 57.7 | 40.3 | 54.1 | 34.5 | 44.3 | 48.8 | 34.8 | 52.7 | 31.2 | 46.0 | 42.4 |
| | *ours* | **43.9** | 59.3 | **41.6** | 56.8 | 34.1 | **45.1** | 48.8 | **35.7** | 56.9 | **31.6** | **47.1** | **43.6** |
| $p = 0$ | Backbone | **43.0** | 60.3 | 40.5 | 56.5 | 34.5 | 34.6 | 32.8 | 26.7 | 42.3 | 28.0 | **47.0** | 32.9 |
| | Finetuning | 36.4 | 42.1 | 32.5 | 43.0 | 29.4 | 42.4 | **49.2** | **36.4** | **60.3** | **32.0** | 36.7 | **44.1** |
| | LoRA | 40.9 | 50.4 | 37.1 | 49.0 | 32.3 | 42.2 | 46.6 | 33.3 | 54.7 | 30.6 | 41.9 | 41.5 |
| | Adapters | 42.2 | 55.1 | 38.4 | 52.8 | 32.9 | 42.3 | 48.3 | 33.6 | 53.5 | 31.5 | 44.3 | 41.9 |
| | *ours* | 41.9 | 58.6 | **39.0** | 54.2 | 33.6 | **43.1** | 48.5 | 35.8 | 56.3 | 31.4 | 45.5 | 43.0 |

We also ran an experiment where **EN-LM-1B** was fine-tuned exclusively on Wikipedia data to assess the impact of this limited distribution on forgetting. Specifically, we fine-tuned **EN-LM-1B** on English Wikipedia for 50,000 steps using a learning rate of $2 \cdot 10^{-4}$, as in Table 10, with the remaining hyperparameters listed in Table 8. This restricted fine-tuning dataset resulted in poor performance across all downstream tasks.

However, when combined with earlier results, this suggests that even when the fine-tuning distribution is only a rough approximation of the original pretraining distribution, strong performance on English downstream tasks can still be maintained.

Table 11. **Impact of training on Wikipedia only.**

| | Forgetting: English | | | | | |
| Method | BeleBele (0) | HellaS (0) | Arc C (5) | CSQA (5) | MMLU (5) | Avg. |
|---|---|---|---|---|---|---|
| Backbone | 43.0 | 60.3 | 40.5 | 56.5 | 34.5 | 47.0 |
| Finetuning | 39.0 | 49.3 | 36.7 | 47.6 | 31.0 | 40.7 |

## D. Math Dataset Experiments

We further fine-tuned the Gemma 2B model (Gemma et al., 2024) on two math-focused datasets—OpenMathInstruct-2 and MetaMathQA—for 50,000 steps using a batch size of $4096 \times 64$ and a peak learning rate of $5 \times 10^{-5}$ across all configurations. For the *Ours* variant, we set the L1 regularization coefficient $\alpha$ to 0.01. The results are presented in Table 12. We observe significant forgetting on English tasks, likely due to the overlap between the mathematical English in the fine-tuning data and general English. The Wikipedia-based English corpus is not sufficient to help the model disambiguate between the two.

*Table 12.* **Maths results**: In **bold** we report the best method, in underline the second best.

| Target | Method | Forgetting: English | | | | | Learning: Target | Task avg. |
| | | BeleBele (0) | HellaS (0) | Arc C (5) | CSQA (5) | MMLU (5) | GSM8K (5) | Avg. score |
|---|---|---|---|---|---|---|---|---|
| *Maths* | Backbone | **47.1** | **71.3** | **47.8** | **64.5** | **39.0** | 20.0 | **53.9** |
| | Finetuning | 43.6 | 64.1 | 35.1 | 56.4 | 31.8 | **58.4** | 46.2 |
| | LoRA | 41.7 | 68.5 | 35.3 | 37.2 | 32.5 | 57.6 | 43.0 |
| | Adapters | 42.8 | 68.6 | 38.1 | 58.7 | 32.4 | 47.6 | 48.1 |
| | *Ours* | 42.9 | 68.2 | 39.2 | 60.8 | 32.0 | 54.4 | 48.6 |

## E. Japanese Dataset Experiments

To assess our approach beyond alphabetic scripts, we conducted additional experiments with Japanese, a language that uses a non-Latin script. Specifically, we augmented Gemma-2B with Japanese data extracted from CommonCrawl, processed using the same pipeline as the other multilingual finetuning datasets. The results are presented in Table 13.

*Table 13.* **Results on Japanese:** In **bold** we report the best method, in underline the second best.

| Target | Method | Forgetting: English | | | | | Learning: Target | | | Task avg. | |
| | | BeleBele (0) | HellaS (0) | Arc C (5) | CSQA (5) | MMLU (5) | BeleBele (0) | CSQA (5) | MMLU (5) | Forgetting (English) | Learning (Target) |
|---|---|---|---|---|---|---|---|---|---|---|---|
| *Japanese* | Backbone | 46.2 | **69.7** | 46.9 | **61.8** | **40.4** | 37.3 | 55.6 | 30.3 | **53.0** | 41.1 |
| | Finetuning | 44.6 | 60.2 | 42.3 | 57.2 | 35.6 | 38.7 | **68.4** | 32.5 | 48.0 | **46.5** |
| | LoRA | 45.7 | 63.3 | 45.5 | 58.7 | 37.4 | 37.3 | 63.0 | 32.4 | 50.1 | 44.2 |
| | Adapters | 46.9 | 65.9 | 46.5 | 60.2 | 38.7 | 37.1 | 63.7 | **32.7** | 51.7 | 44.5 |
| | *Ours* | **47.1** | 66.9 | **48.5** | 61.6 | 39.1 | **39.1** | 64.1 | **32.7** | 52.6 | 45.3 |

## F. Multilingual Dataset Experiments

We explore a multilingual setting where the goal is to learn new languages while preserving performance on previously known ones. We use the Gemma 2 2B model (Team et al., 2024), chosen for its strong multilingual capabilities. The learned languages are Danish, Hungarian, and Slovak; the retained ones are English, French, German, Portuguese, Spanish, and Italian. Training data consists of 90% balanced learned languages and 10% balanced retained languages ($p = 0.1$), approximating the model's original distribution. We compare all methods (with 20% additional parameters, except full fine-tuning) on their ability to learn new languages (Table 15) without degrading performance on retained ones (Table 14). We also evaluate them on FLORES (NLLB Team et al., 2024) (EN → X) in a 5-shot setting.

*Table 14.* **Forgetting results for the multilingual setting:** In **bold** we report the best method, in underline the second best.

| Target | Method | BeleBele (0) | HellaS (0) | Arc C (5) | CSQA (5) | MMLU (5) | FLORES (5) | Avg. |
|---|---|---|---|---|---|---|---|---|
| English | Backbone | **56.3** | **73.9** | **65.6** | **69.1** | **52.3** | x | **63.4** |
| | Finetuning | 46.1 | 59.2 | 40.9 | 53.1 | 33.7 | x | 46.6 |
| | LoRA | 49.3 | 67.8 | 53.8 | 64.0 | 41.3 | x | 55.2 |
| | Adapters | 52.2 | 72.8 | 64.8 | 68.1 | 50.4 | x | 61.7 |
| | *Ours* | 52.8 | 73.3 | 65.4 | 66.6 | 51.6 | x | 61.9 |
| French | Backbone | **48.8** | **60.1** | 56.2 | 58.5 | **44.9** | 43.9 | 52.1 |
| | Finetuning | 45.4 | 50.5 | 37.6 | 42.7 | 30.4 | 34.7 | 40.2 |
| | LoRA | 45.1 | 55.1 | 44.8 | 48.7 | 35.2 | 39.2 | 44.7 |
| | Adapters | 47.6 | 58.7 | 55.4 | 59.3 | 42.6 | 43.6 | 51.2 |
| | *Ours* | **48.8** | 59.7 | **57.1** | **59.9** | 44.0 | **44.1** | **52.3** |
| German | Backbone | **47.6** | **54.0** | 52.5 | 65.0 | 43.7 | **30.8** | 48.9 |
| | Finetuning | 42.4 | 47.5 | 35.3 | 50.8 | 30.8 | 25.2 | 38.7 |
| | LoRA | 44.1 | 50.4 | 44.0 | 53.7 | 35.7 | 26.5 | 42.4 |
| | Adapters | 47.2 | 52.7 | 53.5 | 65.8 | 42.7 | 30.6 | 48.8 |
| | *Ours* | 47.2 | 53.1 | **54.4** | **68.4** | **43.9** | 30.6 | **49.6** |
| Portuguese | Backbone | **46.3** | **58.7** | 56.5 | 68.3 | **45.1** | 44.3 | **53.2** |
| | Finetuning | 40.9 | 50.8 | 37.1 | 51.6 | 30.7 | 37.2 | 41.4 |
| | LoRA | 41.3 | 50.5 | 43.2 | 54.4 | 34.8 | 33.3 | 42.9 |
| | Adapters | 43.9 | 57.4 | 55.3 | **69.5** | 43.1 | **44.5** | 52.3 |
| | *Ours* | 44.4 | 58.1 | **56.7** | 68.6 | 44.4 | 44.0 | 52.7 |
| Spanish | Backbone | **48.1** | **61.2** | **59.2** | x | **45.3** | 25.8 | **47.9** |
| | Finetuning | 41.4 | 51.6 | 40.3 | x | 30.9 | 23.0 | 37.4 |
| | LoRA | 43.8 | 56.1 | 45.2 | x | 36.6 | 24.7 | 41.3 |
| | Adapters | 45.7 | 59.8 | 56.6 | x | 43.7 | 25.8 | 46.3 |
| | *Ours* | 44.3 | 60.6 | 58.2 | x | 44.9 | **26.0** | 46.8 |
| Italian | Backbone | **42.7** | **56.3** | 54.2 | x | **44.4** | 24.8 | **44.5** |
| | Finetuning | 38.4 | 48.5 | 35.1 | x | 30.0 | 21.3 | 34.7 |
| | LoRA | 39.8 | 48.2 | 41.1 | x | 34.8 | 19.4 | 36.7 |
| | Adapters | **42.7** | 54.8 | **54.7** | x | 43.0 | 25.1 | 44.0 |
| | *Ours* | 42.0 | 55.6 | 54.6 | x | 44.0 | **25.6** | 44.4 |

*Table 15.* **Learning results for the multilingual setting:** In **bold** we report the best method, in underline the second best.

| Target | Method | BeleBele (0) | HellaS (0) | Arc C (5) | MMLU (5) | FLORES (5) | Avg. |
|---|---|---|---|---|---|---|---|
| Danish | Backbone | 44.3 | 52.0 | 51.8 | **43.8** | 34.8 | 45.3 |
| | Finetuning | 44.4 | 58.2 | 41.1 | 33.2 | 43.6 | 44.1 |
| | LoRA | 44.6 | **60.4** | 46.9 | 37.1 | **44.2** | 46.6 |
| | Adapters | **46.8** | 58.7 | **54.5** | 43.5 | 43.3 | **49.3** |
| | *Ours* | 44.2 | 56.6 | 53.7 | 43.6 | 41.6 | 48.0 |
| Hungarian | Backbone | 39.7 | 42.6 | 45.6 | 40.2 | 14.0 | 36.4 |
| | Finetuning | 39.4 | **49.4** | 39.0 | 32.2 | **21.5** | 36.3 |
| | LoRA | 33.4 | 35.0 | 33.2 | 29.3 | 20.8 | 30.3 |
| | Adapters | **41.9** | 48.3 | **50.2** | **40.4** | 21.3 | **40.4** |
| | *Ours* | 40.3 | 46.0 | 48.1 | 40.3 | 19.0 | 38.8 |
| Slovak | Backbone | 43.3 | 46.4 | 49.7 | 41.6 | 19.2 | 40.1 |
| | Finetuning | 43.9 | 50.7 | 38.8 | 33.1 | **28.6** | 39.0 |
| | LoRA | 39.1 | 38.2 | 37.3 | 31.2 | 28.4 | 34.8 |
| | Adapters | **44.9** | **50.5** | **50.7** | **41.6** | 28.1 | **43.1** |
| | *Ours* | 43.7 | 48.6 | 49.5 | **41.6** | 25.8 | 41.8 |

# G. Detailed Ablation Results

This section provides the full detailed results corresponding to the ablation study discussed in section 4.4.

*Table 16.* **Ablation: Gating and local loss.**

| | Gate | Local loss | | Forgetting: English | | | | | Learning: French | | | | | Task avg. | |
|---|---|---|---|---|---|---|---|---|---|---|---|---|---|---|---|
| | | $\ell_1$ | CE | BeleBele (0) | HellaS (0) | Arc C (5) | CSQA (5) | MMLU (5) | BeleBele (0) | HellaS (0) | Arc C (5) | CSQA (5) | MMLU (5) | Forget. | Learn. |
| (1) | ∅ | | ✓ | 43.1 | 57.4 | 37.5 | 54.5 | 33.7 | 43.2 | 47.7 | 34.6 | 54.9 | 31.7 | 45.3 | 42.4 |
| (2) | Sigmoid | | ✓ | 41.0 | 57.9 | 40.2 | 53.1 | 34.0 | 43.2 | 47.4 | 36.2 | 54.5 | 31.4 | 45.2 | 42.6 |
| (3) | | ✓ | ✓ | 42.3 | 57.7 | 39.1 | 53.7 | 33.4 | 44.0 | 47.6 | 34.5 | 52.3 | 31.1 | 45.3 | 41.9 |
| (4) | ReLU | | | 43.4 | 59.1 | 39.3 | 55.8 | 34.4 | 44.6 | 48.7 | 34.8 | 54.3 | 31.8 | 46.4 | 42.8 |
| (5) | | | ✓ | 43.9 | 59.3 | 41.6 | 56.8 | 34.1 | 45.1 | 48.8 | 35.7 | 56.9 | 31.6 | 47.1 | 43.6 |
| (6) | Adapter baseline | | | 42.1 | 56.9 | 38.7 | 53.6 | 33.9 | 43.2 | 46.2 | 34.1 | 52.0 | 30.9 | 45.1 | 41.3 |
| (7) | Backbone baseline | | | 43.0 | 60.3 | 40.5 | 56.5 | 34.5 | 34.6 | 32.8 | 26.7 | 42.3 | 28.0 | 47.0 | 32.9 |

*Table 17.* **Ablation: initialization.**

| | | Forgetting: English | | | | | Learning: French | | | | | Task avg. | |
|---|---|---|---|---|---|---|---|---|---|---|---|---|---|
| Gate | Init | BeleBele (0) | HellaS (0) | Arc C (5) | CSQA (5) | MMLU (5) | BeleBele (0) | HellaS (0) | Arc C (5) | CSQA (5) | MMLU (5) | Forget. | Learn. |
| Adapters w/ $\ell_1$-norm | | 43.9 | 57.2 | 39.2 | 53.6 | 33.5 | 42.4 | 46.6 | 34.0 | 53.6 | 31.3 | 45.5 | 41.6 |
| | ✓ | 43.1 | 57.4 | 37.5 | 54.5 | 33.7 | 43.2 | 47.7 | 34.6 | 54.9 | 31.7 | 45.3 | 42.4 |
| Adapters w/ sigmoid | | 42.6 | 57.2 | 38.4 | 56.4 | 33.6 | 43.2 | 46.6 | 32.8 | 50.6 | 30.5 | 45.6 | 40.8 |
| | ✓ | 41.0 | 57.9 | 40.2 | 53.1 | 34.0 | 43.2 | 47.4 | 36.2 | 54.5 | 31.4 | 45.2 | 42.6 |
| Neutral Residues | | 43.0 | 59.2 | 39.9 | 55.8 | 34.7 | 42.0 | 45.7 | 34.3 | 54.9 | 30.9 | 46.5 | 41.6 |
| | ✓ | 43.9 | 59.3 | 41.6 | 56.8 | 34.1 | 45.1 | 48.8 | 35.7 | 56.9 | 31.6 | 47.1 | 43.6 |

*Table 18.* **Ablation:** $\alpha$ controls the strength of the $\ell_1$ local loss.

| | Forgetting: English | | | | | Learning: French | | | | | Task avg. | |
|---|---|---|---|---|---|---|---|---|---|---|---|---|
| $\alpha$ | BeleBele (0) | HellaS (0) | Arc C (5) | CSQA (5) | MMLU (5) | BeleBele (0) | HellaS (0) | Arc C (5) | CSQA (5) | MMLU (5) | Forget. | Learn. |
| 0 | 43.4 | 59.1 | 39.3 | 55.8 | 34.4 | 44.6 | 48.7 | 34.8 | 54.3 | 31.8 | 46.4 | 42.8 |
| 0.01 | 43.9 | 59.3 | 41.6 | 56.8 | 34.1 | 45.1 | 48.8 | 35.7 | 56.9 | 31.6 | 47.1 | 43.6 |
| 0.1 | 42.3 | 59.1 | 38.1 | 56.7 | 34.1 | 44.4 | 48.7 | 34.7 | 55.8 | 31.0 | 46.1 | 42.9 |
| 1 | 43.1 | 59.5 | 39.9 | 56.3 | 34.6 | 43.3 | 48.7 | 34.7 | 53.6 | 31.1 | 46.7 | 42.3 |
| 10 | 42.7 | 59.3 | 39.7 | 57.0 | 34.2 | 44.3 | 48.4 | 35.6 | 55.9 | 31.6 | 46.6 | 43.2 |
| 100 | 43.8 | 59.2 | 38.8 | 55.6 | 34.0 | 45.0 | 48.6 | 34.0 | 56.0 | 31.2 | 46.3 | 43.0 |

*Table 19.* **Trade-offs with different learning rates.**

| Method | LR | Forgetting: English | | | | | Learning: French | | | | | Task avg. | |
|---|---|---|---|---|---|---|---|---|---|---|---|---|---|
| | | *BeleBele (0)* | *HellaS (0)* | *Arc C (5)* | *CSQA (5)* | *MMLU (5)* | *BeleBele (0)* | *HellaS (0)* | *Arc C (5)* | *CSQA (5)* | *MMLU (5)* | *Forget.* | *Learn.* |
| Backbone | 0 | 43.0 | 60.3 | 40.5 | 56.5 | 34.5 | 34.6 | 32.8 | 26.7 | 42.3 | 28.0 | 47.0 | 32.9 |
| Fine-tuning | $2 \cdot 10^{-5}$ | 41.9 | 58.3 | 37.9 | 54.5 | 33.6 | 38.3 | 37.2 | 29.5 | 47.5 | 29.1 | 45.2 | 36.3 |
| | $5 \cdot 10^{-5}$ | 41.8 | 55.8 | 37.5 | 53.7 | 33.1 | 41.7 | 44.1 | 34.0 | 54.8 | 30.8 | 44.4 | 41.1 |
| | $2 \cdot 10^{-4}$ | 37.9 | 45.5 | 34.8 | 45.5 | 30.9 | 42.2 | 48.6 | 35.3 | 60.3 | 31.5 | 38.9 | 43.6 |
| | $5 \cdot 10^{-4}$ | 36.9 | 40.4 | 32.1 | 41.1 | 29.5 | 42.8 | 47.6 | 34.0 | 57.3 | 30.9 | 36.0 | 42.5 |
| LoRA | $2 \cdot 10^{-5}$ | 42.3 | 54.7 | 38.9 | 51.5 | 32.9 | 40.9 | 44.4 | 32.4 | 54.7 | 30.1 | 44.1 | 40.5 |
| | $5 \cdot 10^{-5}$ | 42.0 | 53.1 | 38.1 | 52.3 | 32.8 | 42.4 | 46.2 | 32.7 | 54.2 | 30.7 | 43.7 | 41.2 |
| | $2 \cdot 10^{-4}$ | 41.3 | 51.1 | 36.1 | 48.4 | 32.4 | 42.1 | 47.3 | 33.8 | 53.9 | 31.2 | 41.9 | 41.7 |
| | $5 \cdot 10^{-4}$ | 41.1 | 50.8 | 36.6 | 47.0 | 32.3 | 44.7 | 48.0 | 34.4 | 54.8 | 31.0 | 41.6 | 42.6 |
| Vanilla adapters | $2 \cdot 10^{-5}$ | 43.0 | 56.9 | 38.7 | 52.8 | 34.0 | 41.7 | 44.6 | 33.3 | 54.3 | 30.3 | 45.1 | 40.8 |
| | $5 \cdot 10^{-5}$ | 42.1 | 56.9 | 38.7 | 53.6 | 33.9 | 43.2 | 46.2 | 34.1 | 52.0 | 30.9 | 45.1 | 41.3 |
| | $2 \cdot 10^{-4}$ | 43.1 | 57.7 | 40.3 | 54.1 | 34.5 | 44.3 | 48.8 | 34.8 | 52.7 | 31.2 | 46.0 | 42.4 |
| | $5 \cdot 10^{-4}$ | 42.3 | 57.9 | 39.2 | 53.8 | 33.6 | 45.2 | 48.6 | 36.2 | 56.2 | 31.5 | 45.4 | 43.5 |
| Neutral residues | $2 \cdot 10^{-5}$ | 42.6 | 59.1 | 40.8 | 55.5 | 33.9 | 44.9 | 47.4 | 35.6 | 54.6 | 31.9 | 46.4 | 42.9 |
| | $5 \cdot 10^{-5}$ | 43.9 | 59.3 | 41.6 | 56.8 | 34.1 | 45.1 | 48.8 | 35.7 | 56.9 | 31.6 | 47.1 | 43.6 |
| | $2 \cdot 10^{-4}$ | 42.7 | 59.1 | 40.7 | 55.0 | 34.5 | 43.6 | 49.2 | 35.2 | 53.1 | 31.6 | 46.4 | 42.5 |
| | $5 \cdot 10^{-4}$ | 42.8 | 59.3 | 39.4 | 57.2 | 34.2 | 44.0 | 49.1 | 35.2 | 55.2 | 31.2 | 46.6 | 43.0 |

