# OpenReview forum: "Neutral residues: revisiting adapters for model extension"
_ICML.cc/2025/Conference — ICML 2025 poster_

### Official Review · Reviewer_CFSi · 2025-03-13

**Overall Recommendation:** 3

**Summary:**

This paper presents Neutral Residues, a method for extending large language models (LLMs) to new domains while mitigating catastrophic forgetting. The proposed method builds upon adapter-based techniques (to add extra capacity to the model), introducing architectural modifications with parallel gated adapters, regularizes adapter outputs to produce near-zero activations on the original domain, and implements a low-variance initialization strategy (based on He's initiallization) to improve adaptation stability.
These techniques aim to optimize the trade-off between learning new knowledge and retaining previous capabilities.
The paper primarily evaluates its approach in the context of multilingual adaptation (from English to other languages), demonstrating that Neutral Residues effectively achieve the best trade-off between minimizing forgetting and improving the target task compared to finetuning, LoRA, and standard adapter baselines.

**Claims And Evidence:**

**Main claim**: Neutral Residues improves the trade-off between learning and forgetting.
- This claim is supported by findings in Tables 2,3 showing comparable performance in English to the pretrained model and high downstream task performance in target languages (e.g., MMLU, ARC, HellaSwag)

The effect of the proposed components, including ReLU-based gating, Low-variance initialization, and local loss, are validated with ablation studies shown in tables 4, 5, and 6, mostly with English-French pairs.

**Sufficiency of evidence**: While the results are clear and supportive of the claim, it needs more evidence from diverse sets of language pairs and potentially other domains and tasks. In particular,
* Only the English-pretrained model is used. Also, the target languages should be more diverse with the language taxonomy
* Ablation studies and analyses are conducted with only English-French pairs, which are not sufficient for strong claims.
* Broader applications (beyond multilingual transfer) should be discussed and evaluated for stronger claims.
* l1 loss: How does this loss work if the two languages are close, e.g., English and German? May this loss hurt the learning of the target domain?
* Forgetting: Can we evaluate the models on the pretrained data distribution where the pretrained model probably works the best?
* Version of LoRA and Adapters: There are subsequent works of these methods that show advantages in domain adaptation, e.g., DoRA (https://arxiv.org/abs/2402.09353), QLoRA (https://arxiv.org/abs/2305.14314).

**Essential References Not Discussed:**

No

**Experimental Designs Or Analyses:**

Yes, I checked and validated all sections in the experiment, from 4.2 (preliminary analysis) to the ablation studies.

**Methods And Evaluation Criteria:**

**Method**:
The method is sound, backed by intuitions, empirical observations, and evidence.
These assumptions are somewhat validated with empirical validation (though only on a pair of multilingual transfer).

**Evaluation**:
The experiments are well-structured, using Gemma-2B, EN-LM-1B as backbone models and comprehensive benchmarks (perplexity + downstream tasks)
The ablation studies for gating strategies, initialization, and loss functions are sensitive.

*Limitations*:
 * The datasets are primarily multilingual, so performance on other knowledge domains is unclear.
 * Only one main backbone architecture (transformer) is tested, so results may not generalize to others, like vision models.

**Other Comments Or Suggestions:**

Please see the comments above, including additional tasks and languages.

**Other Strengths And Weaknesses:**

**Strengths**
- The studied problem is interesting and important
- The problem and method are well-articulated and presented with backed evidence.
- The experiment designs are thorough for understanding components of the method.

**Weaknesses**:
- Scalability: The method often adds 20% extra parameters, which significantly increases inference costs.
- Results focus on multilingual adaptation, so effectiveness on non-linguistic tasks (e.g., code, biology) is unknown.
- The paper needs more extensive evaluation for stronger claims.

**Questions For Authors:**

Together with the questions above, I have several questions:

1. Motivation: Can you elaborate or provide a simple example for which LoRA doesn't add capacity and significantly affects the downstream performance?
2. For the local loss: Why do you choose l1 (sparsity) over other losses, e.g., L-2 or L-infinity (for zero out all), as the L1 is generally considered harder for optimization?
3. For LoRA, can you point out the results between applying LoRA on FFN and attention for the tasks?

**Relation To Broader Scientific Literature:**

This paper contributes to addressing catastrophic forgetting in model extension, which is highly relevant for the current research of efficient fine-tuning of large foundational models, given the growing cost of retraining foundation models.

**Theoretical Claims:**

No claim

---

> ### Author Rebuttal · Authors · 2025-04-01
>
> First, we would like to thank the reviewer for their feedback on our paper.
> 1. **Only the English-pretrained model is used. Also, the target languages should be more diverse with the language taxonomy.**
>
> Our experiments are also conducted on Gemma, which was trained on a small amount of multilingual data (although the goal was not to reach SOTA performance on multilingual tasks). In particular, it obtains non-random performance on the multilingual evals we consider.
>
> We also ran new experiments by adding Japanese to Gemma-2B. We extract data from CommonCrawl, and process it with the same pipeline as the rest of the multilingual finetuning datasets. The setting is the same as in table 3.
>
> | Method |Forgetting (EN) | Learning (JA) |
> |-|-|-|
> | Base |53.0|38.8|
> | Finetuning |48.0|46.5|
> |Lora|50.1|44.2|
> |Adapters|51.7|44.5|
> |Ours|52.6|45.3|
>
>
> 2. **Ablation studies and analyses are conducted with only English-French pairs**
>
> Unfortunately, due to compute constraints, we could not run ablations on all language pairs.
>
> 3. **Broader applications (beyond multilingual transfer) should be discussed and evaluated for stronger claims**
>
> We trained gemma 2B on math datasets ([OpenMathInstruct-2](https://huggingface.co/datasets/nvidia/OpenMathInstruct-2) and [MetaMathQA](https://huggingface.co/datasets/meta-math/MetaMathQA))  over 50000 steps with a batch of 4096*64, lr=5e-5 for all settings and the coefficient of the L1 norm for *ours* remains 0.01.
>
> | Method | EN | GSM8K|
> |-|-|-|
> | Base | 53.7 | 20 |
> | Finetuning | 45.5 | 58.4 |
> | Lora  | 43.7 | 57.6 |
> | Adapters  | 48.1 | 47.6 |
> | Ours | 48.1 | 54.4 |
>
> 4. **L1 loss: How does this loss work if the two languages are close, e.g., English and German? May this loss hurt the learning of the target domain?**
>
> We conducted experiments by adding German to EN-LM-1B. We extract data from CommonCrawl, and process it with the same pipeline as the rest of the multilingual finetuning datasets. The model is trained with 20% of extra learnable parameters and the remaining hyperparameters are similar to table 8. Here are the results.
>
> | Method |Forgetting (EN) | Learning (DE) |
> |-|-|-|
> | Base |47.0|32.0|
> | Finetuning |38.4|44.1|
> |Lora|43.0|41.2|
> |Adapters|44.5|41.9|
> |Ours|46.7|41.0|
>
> 5. **Forgetting: Can we evaluate the models on the pretrained data distribution where the pretrained model probably works the best?**
>
> Thanks for the suggestion. As we do not have the pretraining distribution for Gemma, we can only run this for the LM-EN-1B model. We compute the perplexity on the pretraining distribution for models from table 7 (lr=2e-4 for all):
>
> | Method |Forgetting (EN) |
> |-|-|
> | Base |0.781|
> | Finetuning |0.937|
> |Lora|0.885|
> |Adapters|0.821|
> |Ours|0.796|
>
> As we can observe, this does not change the conclusion, compared to using PubMed to measure forgetting. We will add these results to the appendix.
>
> 6. **Only one main backbone architecture (transformer) is tested, so results may not generalize to others, like vision models.**
>
> Thanks for the suggestion: exploring architecture different from transformer would be interesting, but due to time constraints, we leave it for future work.
>
> 7. **Motivation: Can you elaborate or provide a simple example for which LoRA doesn't add capacity and significantly affects the downstream performance**
>
> Table 8 shows that downstream performance on French with Lora was significantly lower than that achieved with finetuning when training EN-LM-1B on French. This was particularly noticeable when only a few extra learnable parameters were added.
>
> 8. **Why do you choose L1 (sparsity) over other losses, e.g., L-2 or L-infinity [...]?**
>
> When using the L2 loss, the model still exhibited significant forgetting. This occurs because, as the outputs of the residual blocks approach zero, the gradients of their L2 loss become small, making them insufficient to effectively drive the outputs toward zero. In contrast, the L1 loss maintains constant gradients, enabling it to push outputs closer to zero more effectively. We also explored a normalized L2 loss, which mitigates the issue of small gradients. However, preliminary experiments did not show significant improvements, so we ultimately chose the L1 loss.
>
> 9. **Can you point out the results between applying LoRA on FFN and attention for the tasks?**
>
> We trained EN-LM-1B on French data using the same hyperparameter settings as in Table 8, with an additional 20% of learnable weights. LoRA was applied either to the FFN, the attention layers, or both. Our results indicate that applying LoRA only to the FFN is better than applying it solely to the attention layers and achieves similar performance to using it on both.
>
> | Method | Tasks  |||  Perplexity ||
> |-|-|-|-|-|-|
> |  | EN | FR || EN| FR |
> |Attn|43.4|39.5|| 0.710 | 0.857|
> |FFN|43.4|40.9|| 0.730 | 0.819|
> |Both|43.4|41.0|| 0.725| 0.824|

---

> > ### Comment · Reviewer_CFSi · 2025-04-07
> >
> > Thanks for the response.
> >
> > I have checked and kept my initial recommendation.
> >
> > Best,

---

### Official Review · Reviewer_Vwee · 2025-03-14

**Overall Recommendation:** 2

**Summary:**

Extending a pre-trained large language model (LLM) to a new domain/language is challenging. It is known that such model extension often encounter a trade-off, between performing well on the new domain/language vs degrading performance on the original domain/language. This paper addresses such the problem by revising adaptors. In the experiments, they report perplexity to measure how the extended model performs well on the targeted domain/language, as well as model performance in the downstream tasks such as question answering. Their carefully designed experiment and its extensive results contribute to underscore several critical factors such as data, architecture, and training, and initialization. These findings would be helpful for readers.

**Claims And Evidence:**

Experimental results  in Table 1 show that the proposed approach work well while balancing the existing domain/language (English) and the new domain/language (French). They deliberately explored the hyperparameter in the preliminary experiments.

**Essential References Not Discussed:**

N/A

**Experimental Designs Or Analyses:**

As another backbone model option, you could also use a multilingual LLM to assess the effectiveness of your proposed approach. I am curious to see how sensitive the proposed approach is to hyperparameter selection in such a complexed multilingual setting.

In the experiments, the targeted languages are mostly European languages with shared alphabetical scripts. Have you ever tried to employ other languages with different scripts such as Arabic and Chinese?

**Methods And Evaluation Criteria:**

The results in the downstream tasks do not seem to give consistent improvement across different tasks, while the proposed approach achieves the best or equivalent performance against the well-known approaches such as finetuning, LoRA, and Adapters.

I was wondering how robust the proposed approach as the number of new domain/language increases. Have you ever tried out any multilingual settings including more than 2 languages in total?

**Other Comments Or Suggestions:**

Please see my comments in the sections above.

**Other Strengths And Weaknesses:**

Please see my comments in the sections above.

**Questions For Authors:**

Please see my questions in the sections above.

**Relation To Broader Scientific Literature:**

The trade-off issue, between model performance in the new domain vs in the original domain, is very crucial when extending the LLMs. This approach sheds another light on this direction, achieving slightly better performance against the other techniques such as finetuning, LoRA, and Adapters.

**Theoretical Claims:**

N/A

---

> ### Author Rebuttal · Authors · 2025-04-01
>
> First, we would like to thank the reviewer for their feedback on our paper.
>
> 1. **Have you ever tried out any multilingual settings including more than 2 languages in total?**
>
> We conducted new experiments by adding French, Danish, Hungarian, and Slovak simultaneously to Gemma 2B. The training is done with the same hyperparameter setting as in Table 3, over 100,000 steps, using 10% English data and the four languages in equal parts. We report the mean task performance for each language.
>
> | Method | EN | FR | DA | HU |SK|
> |-|-|-|-|-|-|
> | Base | 53.0 | 44.2 | 37.6 |33.1 | 34.6|
> | Finetuning | 48.6 | 47.3 |43.3 |38.3| 40.7|
> | Lora  | 50.3 | 45.8 |41.5 |37.2|38.7|
> | Adapters  | 50.9 | 44.9 |41.4 |37.1|38.2 |
> | Ours | 52.4 | 46.0 |41.2 |36.7|39.1|
>
> 2. **As another backbone model option, you could also use a multilingual LLM to assess the effectiveness of your proposed approach. I am curious to see how sensitive the proposed approach is to hyperparameter selection in such a complex multilingual setting.**
>
> Gemma was trained on a small amount of multilingual data (although the goal was not to reach SOTA performance on multilingual tasks). In particular, it obtains non-random performance on the multilingual evals we consider. Do you mean to assess the effectiveness of our approach to preserve performance on multilingual tasks?
>
> 3. **Have you ever tried to employ other languages with different scripts [...] ?**
>
> We conducted experiments by adding Japanese to Gemma-2B. We extract data from CommonCrawl, and process it with the same pipeline as the rest of the multilingual finetuning datasets. The hyperparamter setting is the same as in table 3.
>
> | Method |Forgetting (EN) | Learning (JA) |
> |-|-|-|
> | Base |53.0|38.8|
> | Finetuning |48.0|46.5|
> |Lora|50.1|44.2|
> |Adapters|51.7|44.5|
> |Ours|52.6|45.3|

---

### Official Review · Reviewer_5QgN · 2025-03-16

**Overall Recommendation:** 2

**Summary:**

This paper addresses the challenge of extending a pretrained large language model to a new domain (e.g., a new language) without catastrophic forgetting of the original domain. The authors propose “neutral residues,” a method that adds adapter layers to the model and trains them such that their outputs are near-zero for original-domain inputs​. Overall, the idea is intuitive and the results are promising, but some limitations in scope and novelty lead me to lean towards a weak rejection of this paper in its current form.

**Claims And Evidence:**

The primary claim is that neutral residue adapters enable superior domain adaptation compared to existing methods by preserving original task performance while learning the new domain​. The evidence comes from experiments: the authors show that a model with neutral residue adapters achieves lower perplexity on English (original domain) and comparable or better performance on the new language, versus baselines that either forget English or underperform in the new language. They also evaluate on benchmark QA and knowledge tasks (ARC, HellaSwag, MMLU, etc.) in both English and the target language, where the adapted model maintains strong English accuracy while gaining new-language capability. The evidence is credible that neutral residues work well in the tested scenario. However, it is mostly limited to one setting (one original model and one target domain), so the generality of the claim (to other domains or models) is not fully proven.

**Essential References Not Discussed:**

no

**Ethical Review Concerns:**

n.a.

**Experimental Designs Or Analyses:**

The experimental design is straightforward and solid for the scenario considered. The authors take a state-of-the-art English language model and adapt it to a single target language (the paper implies French as the target, given “FR” in results). They compare multiple methods under the same conditions (same base model, same new data). This controlled setup makes the comparisons fair. They evaluate on multiple axes: perplexity on two domains and accuracy on five benchmark tasks, which provides a holistic view of performance.

One commendable aspect is the evaluation of different proportions of original data mixed during training​. In Table 1, they vary the fraction of English data in the adapter training and find ~10% is a good trade-off, which empirically supports their data mixing strategy. This kind of ablation is useful. They also presumably keep the original model’s weights frozen for all adapter methods to ensure comparability (since that’s how adapters typically operate).

A minor critique is that the experiments focus on only one new domain (language). We don’t see results for, say, adapting to a second language or a different domain (like adapting an English model to scientific text or code). So it’s unclear if the approach consistently works in other settings or if any domain-specific tuning was needed. Additionally, while they beat baselines, it would help to understand which component of their approach contributes most – e.g., is it the data replay that helps more, or the architectural constraint of near-zero outputs? The paper doesn’t fully separate these; an ablation where they train adapters without the near-zero constraint (just with data mixing) could isolate the effect. The analysis currently attributes the success to the combination of all changes. Despite these points, the overall analysis of forgetting vs. learning is thorough for the given setting, and the improvements are clearly demonstrated (often neutral residues get the best of both worlds: low new-language perplexity with minimal English degradation).

**Methods And Evaluation Criteria:**

Method: Neutral residues modify the standard adapter approach in three aspects (as hinted by the paper): data, architecture, and training procedure. Architecturally, they insert adapter layers (small learned modules) and initialize/train them such that on original-domain inputs their contribution is nearly zero, hence “neutral”​. In training, they mix a small amount of original-domain data into the fine-tuning of adapters to explicitly guard against forgetting​. This data mixing ensures the adapter doesn’t drift too far from the original distribution. The training procedure likely also involves a special initialization (the paper references prior work that initializing adapters to near-identity is important) so that initially the model’s behavior is unchanged.

Evaluation: They use a two-step evaluation: (1) Perplexity on held-out English vs. new-language text to quantify forgetting vs. learning​. Lower perplexity is better; an ideal adaptation would keep English perplexity low (close to the original model) while improving new-language perplexity. (2) Downstream tasks performance on standard benchmarks in both languages​, such as question answering and commonsense reasoning tasks. They compare against fine-tuning the whole model, Low-Rank Adaptation (LoRA), and vanilla adapters. The use of perplexity and a diverse suite of tasks is appropriate, covering both intrinsic performance and extrinsic task efficacy. The criteria focus on the trade-off curve between new task gain and original task loss – a key aspect of this problem. The paper’s results highlight, for instance, that neutral residues achieve a better balance on this trade-off curve, outperforming baselines at equivalent points​.

**Other Comments Or Suggestions:**

1. Ablation of Components: As mentioned, an ablation study would strengthen the paper. For example, train an adapter with data replay but without enforcing neutral initialization, and vice versa, to see which contributes more. This would inform if the “three angles” (data, architecture, training) are all necessary.
2. Generality to Multiple Domains: If space permits (or in future work), it’d be great to test adding two new domains sequentially (to simulate continual learning). Does the first adapter remain neutral while a second is added? Perhaps stacking adapters for each new domain could be explored.
3. Analysis of Forgetting vs. Capacity: The paper could discuss an interesting insight: fine-tuning fails because it has to shove new knowledge into existing weights, whereas their adapter adds new weights. It might be worth highlighting how their approach relates to the idea of “soft modularity” – you’re effectively modularizing knowledge (base model for old stuff, adapter for new). This perspective could resonate with the continual learning community.

**Other Strengths And Weaknesses:**

Strengths:
   1. Addresses a crucial problem: As models get larger, being able to extend them without full retraining is very important (for efficiency and sustainability). This work directly tackles that by enabling domain expansion at low cost.
    2. Simplicity and Elegance: The idea of making adapter outputs neutral on old data is simple yet clever. It doesn’t require a complex loss function besides possibly using some original data. It’s an elegant tweak that has a big effect on forgetting.
    3. Empirical Performance: The method shows clear empirical gains. It outperforms strong baselines (fine-tuning, LoRA) in preserving original performance while learning new language​. The results are consistent across both perplexity and benchmark evaluations, adding credibility.
    4. Comprehensive Evaluation: I appreciate that they evaluated on multiple metrics (perplexity and QA benchmarks) and considered different proportions of replay data. This gives a well-rounded view of the method’s behavior.
    5. Efficiency: Using adapters keeps the number of trained parameters small (adapters typically add only ~3-5% of parameters). So, the approach is computationally efficient – an important practical strength, aligned with the paper’s motivation of avoiding huge retraining costs.

Weaknesses:
    1. Incremental Novelty: The method is essentially an improved adapter training recipe. Adapters and data replay are existing ideas; the main novelty is the “near-zero output” constraint. While useful, it’s not a huge conceptual leap beyond prior work. Some might view it as an incremental improvement rather than a fundamentally new method.
    2. Limited Experiment Scope: The experiments are mostly on adapting to one new language. It’s unclear how the method performs for different kinds of domain shifts (e.g., style or topic changes, or adding multiple new domains sequentially). The paper would be stronger if it demonstrated success in more than one scenario. Right now, it’s possible the approach was tuned specifically for the one case.
    3. Lack of Theoretical Insight: There isn’t a deeper analysis of why forcing near-zero outputs is the best way to retain knowledge. For instance, could this be at odds with learning the new task (since you’re constraining the model)? The paper doesn’t theoretically guarantee anything about forgetting, so one must trust the empirical results. A bit more explanation (even qualitative) of how neutral the adapters remained (did they truly output near zeros on English inputs?) would help understand the mechanism.

**Questions For Authors:**

1. Enforcing Neutral Outputs: Could you elaborate on how you ensure the adapter outputs are near-zero for the original domain? Do you initialize the adapter’s final linear layer to zero weights (so initial output is zero) and then rely on the presence of original data during training to keep it low? Or do you use an explicit loss term that penalizes any deviation on English examples? Clarifying this would help understand how “neutrality” is maintained throughout training.
2. Generality to Other Domains: Have you tried applying neutral residues to a different kind of domain shift, such as adapting an LLM trained on general text to a domain like legal texts or code? If not, what do you expect – would the method work equally well, and would you need any adjustments? Any insight into this would tell us about the scope of applicability.

**Relation To Broader Scientific Literature:**

This work is related to continual learning, domain adaptation, and parameter-efficient fine-tuning of large models. The authors do well to situate it: they cite the original adapters paper (Houlsby et al., 2019) and other PEFT (Parameter-Efficient Fine-Tuning) methods​. They also connect to catastrophic forgetting literature, citing classic works (McCloskey & Cohen, 1989; French, 1999) and more recent ones like Elastic Weight Consolidation (Kirkpatrick et al., 2017)​.

**Theoretical Claims:**

There are no new formal theoretical claims in this work. The paper is largely empirical. It builds on the known concept that adding capacity (via adapters) can mitigate catastrophic forgetting (since fine-tuning with no new parameters has an inherent capacity trade-off​). The authors reference the theory of catastrophic forgetting from continual learning literature and conceptually argue that, because fine-tuning and LoRA do not add capacity, they are “inherently limited” and will eventually forget earlier knowledge​. Neutral residues, by adding extra parameters, avoid this limitation. However, this is an intuitive argument rather than a new theory. The paper does not provide theoretical guarantees (e.g., no formal proof that adapter outputs remain zero or that forgetting is bounded). The contribution is primarily a technique and its empirical validation.

---

> ### Author Rebuttal · Authors · 2025-04-01
>
> First, we would like to thank the reviewer for their feedback on our paper.
> 1. **Limited Experiment Scope: The experiments are mostly on adapting to one new language.**
>
> Exploring sequentially adding multiple domains would be interesting, but due to time constraints, we leave it for future work. However, we conducted new experiments by adding French, Danish, Hungarian, and Slovak simultaneously to Gemma 2B, as also recommended by the reviewer CFSi: “*Have you ever tried out any multilingual settings including more than 2 languages in total ?* ”. The training is done with the same hyperparameter setting as in Table 3, over 100,000 steps, using 10% English data (wikipedia dataset) and the four languages in equal parts. We report the mean task performance for each language.
>
> | Method | EN | FR | DA | HU |SK|
> |-|-|-|-|-|-|
> | Base | 53.0 | 44.2 | 37.6 |33.1 | 34.6|
> | Finetuning | 48.6 | 47.3 |43.3 |38.3| 40.7|
> | Lora  | 50.3 | 45.8 |41.5 |37.2|38.7|
> | Adapters  | 50.9 | 44.9 |41.4 |37.1|38.2 |
> | Ours | 52.4 | 46.0 |41.2 |36.7|39.1|
>
> We also trained gemma 2B on math datasets ([OpenMathInstruct-2](https://huggingface.co/datasets/nvidia/OpenMathInstruct-2) and [MetaMathQA](https://huggingface.co/datasets/meta-math/MetaMathQA))  over 50000 steps with a batch of 4096*64, lr max of 5e-5 for all settings and the coefficient of the L1 norm for *ours* remains 0.01.
>
> | Method | EN | GSM8K|
> |-|-|-|
> | Base | 53.7 | 20 |
> | Finetuning | 45.5 | 58.4 |
> | Lora  | 43.7 | 57.6 |
> | Adapters  | 48.1 | 47.6 |
> | Ours | 48.1 | 54.4 |
>
> 2. **Ablation of Components: [...] train an adapter with data replay but without enforcing neutral initialization, and vice versa**
>
> To demonstrate the significance of data, architecture, and training in mitigating forgetting during learning, we conducted several ablation studies:
>
> - Impact of Mixed Data Distribution (Section 4.2, Table 9 in the Appendix): We analyzed the effect of training with a mixed data distribution across all settings, emphasizing the importance of maintaining a distribution that approximates the pretraining distribution.
>
> - Ablation of Gating and Local Loss (Table 4): We investigated various gating mechanisms and their training approaches, demonstrating their role in distinguishing data similar to the pretraining distribution from newly learned data.
>
> - Initialization Ablation (Table 5): We compared adapters trained with and without our initialization, using both L1 loss and standard cross-entropy loss (see Section 3, “Sigmoid activation with cross-entropy”).
>
> Taken together, these experiments highlight that all three factors—data, architecture, and training—are essential.
>
> Combining the ablation studies, we replicate the reviewer's suggestion by comparing neutral residues with data replay without neutral initialization and vice versa instead of adapters. Below are results on EN-LM-1B trained on French data:
>
> | Method | Tasks |||  Perplexity ||
> |-|-|-|-|-|-|
> |  | EN | FR || EN| FR |
> | data & no init | 46.5 | 38.4  ||0.673|0.818|
> | no data & init |46.0 | 40.7||0.684|0.789|
>
> This highlight the importance the initialization to reduce the forgetting through the training without hurting the learning.
>
> We agree with the reviewer that these ablations were underemphasized and adding experiments, such as the previous one on adapters, will strengthen the studies. These will be included in the final version.
>
> 3. **Lack of Theoretical Insight**
>
> Using the EN-LM-1B model finetuned on French, we compute the ratio of the L1 norms of the output of adapters and the output of the backbone MLP. We compare vanilla adapters (A) to neutral residues (B), to illustrate the impact of our method. We average over 2.6M tokens:
>
> **On English Pubmed**
> | Layer | A | B |
> |-|-|-|
> | 3 | 0.208 | 0.012 |
> | 7 | 0.209 | 0.011 |
> | 11 | 0.375 | 0.005 |
> | 15 | 0.266 | 0.002 |
>
> **On French valid set**
> | Layer | A | B |
> |-|-|-|
> | 3 | 0.557 | 0.547 |
> | 7 | 0.379 | 0.412 |
> | 11| 0.703 | 0.730 |
> | 15 | 1.578 | 1.453 |
>
> This experiment reveals that gating minimizes residual outputs in English to reduce forgetting.
>
> 4. **Analysis of Forgetting vs. Capacity**
>
> Thanks for the suggestion, we will add a discussion in the final version of the paper.
>
> 5. **Enforcing Neutral Outputs: Could you elaborate on how you ensure the adapter outputs are near-zero for the original domain ?**
>
> In fact as described in section 3 part “Low-variance initialization” we initialize the adapter’s final linear layer to zero so that initial output is zero. Then during training the L1 loss is applied for data approximating the pretraining distribution to help maintain the output of the new blocks near zero for those data.

---

### Official Review · Reviewer_JvbR · 2025-03-20

**Overall Recommendation:** 3

**Summary:**

Authors explore a set of techniques and strategies for reducing the compute needed to train an already trained LLM for a new task or language. These include the ratio of the training data to the pretraining data, the architecture of the newly added modules (ffd or multi-head attention), the way that they are added to the network (sequential or parallel), the effectiveness of adding the gating mechanism to the new modules, the types of gating mechanism (relu or sigmoid), the ways to train the gating mechanism, and the initialization of the added modules. In some cases the topics have been already explored in previous studies (e.g., training data, the architecture of the newly added modules, the way that they are added, etc), and in some other cases the topics are drawn from closely related research areas (e.g., types of the gating mechanism).

The techniques mentioned above are evaluated in a set of experiments and in most cases are shown to be marginally effective.

**Claims And Evidence:**

Yes

**Essential References Not Discussed:**

None

**Experimental Designs Or Analyses:**

Yes

**Methods And Evaluation Criteria:**

Yes

**Other Comments Or Suggestions:**

None

**Other Strengths And Weaknesses:**

**Strengths:**

-  The discussion is detailed, and in some cases the arguments are insightful.
-  The experiments are convincing and insightful.
-  The paper is easy to read.

**Weaknesses:**

- To my understanding, all the techniques and strategies discussed in the paper are drawn from previous studies. It is still nice to see all of them in one paper, but it also makes me a bit reluctant to recommend it as an icml paper. I am not sure.
- There is not point in reporting ablation studies in average performance. Ablation studies are reported for detailed comparison across tasks.
- The improvements (Table 3) in my opinion are not significant.

All in all this is an old style deep learning paper that discusses a set of techniques that show that empirically work better than the alternatives.

**Questions For Authors:**

None

**Relation To Broader Scientific Literature:**

The study mostly discusses already existing techniques

**Theoretical Claims:**

N/A

---

> ### Author Rebuttal · Authors · 2025-04-01
>
> First, we would like to thank the reviewer for their feedback on our paper.
>
> 1. **To my understanding, all the techniques and strategies discussed in the paper are drawn from previous studies**.
>
> Taken individually, some techniques used in our paper were indeed proposed in previous work. However, a key insight of our paper is the interplay between these different modifications of adapters, and the fact that they cannot be studied effectively independently. Moreover, we believe that some of our proposed modifications are original, such as constraining the output of an adapter module to zero with an L1 norm.
>
> 2. **There is no point in reporting ablation studies in average performance.**
>
> Thanks for the suggestion, we will detailed results in the appendix of the final version of the paper.

---

### Decision · Program_Chairs · 2025-05-01

**Decision:**

Accept (poster)

**Comment:**

This paper proposes "Neutral Residues," an adapter-based method aimed at extending LLMs to new domains/languages while mitigating catastrophic forgetting of the original task. The approach combines parallel gated adapters, data replay, L1 regularization for neutrality on original data, and specific initialization. Initial reviews were split between weak accept and weak reject, but the authors provided a thorough rebuttal with significant new experiments (multi-language, non-European script, math domain) and analyses addressing key concerns about scope and component contributions. One reviewer explicitly raised their score to weak accept post-rebuttal, resulting in a majority leaning towards acceptance. While some concerns about incremental novelty and scalability remain, the demonstrated empirical success in balancing learning and forgetting across several settings makes this a potentially useful contribution. The only rejecting reviewer (Vwee) initially provided a short review with few general questions raised. Those seem to have been addressed by the authors, yet the reviewer did not comment on their responses with additional questions, simply acknowledged them.

**Strengths:**
* Addresses a crucial and timely problem: Efficiently extending LLMs without catastrophic forgetting (5QgN, Vwee, CFSi).
* Demonstrates clear empirical gains and achieves a good trade-off between preserving original domain performance and learning the new domain/language across multiple experiments and metrics (JvbR, 5QgN, Vwee, CFSi).
* Methodology is generally well-explained, and the experiments, including preliminary studies and ablations, are relatively thorough and convincing within the tested scope (JvbR, 5QgN, Vwee, CFSi).
* The core idea of enforcing neutrality via L1 loss on adapter outputs for original domain data is intuitive and elegant (5QgN).
* The paper is generally well-written and easy to follow (JvbR, CFSi).

**Points for improvement:**
* **Limited Experimental Scope/Generality:** While the authors added new experiments in the rebuttal (multi-language, Japanese, math), the evaluation still primarily uses two base models and hasn't explored sequential adaptation, diverse model architectures (beyond transformers), or a wider range of domain types. (Initial concern: 5QgN, Vwee, CFSi; Rebuttal: Added experiments; Acknowledgment: Partially addressed for 5QgN, acknowledged by Vwee/CFSi).
* **Incremental Novelty:** The approach combines several existing techniques (adapters, gating, data replay, specific initializations). The main novelty lies in their specific combination and the L1 regularization strategy, which some reviewers felt was an incremental improvement. (Initial concern: JvbR, 5QgN; Rebuttal: Argued interplay is novel; Acknowledgment: Acknowledged by JvbR/5QgN, likely remains a valid point).
* **Insufficient Ablation/Analysis:** Although new ablations and analyses were provided (component comparison, L1 norm analysis, pretraining data perplexity), some desire for deeper theoretical insight or ablations across more diverse settings might remain. (Initial concern: JvbR, 5QgN, CFSi; Rebuttal: Added analysis/experiments, pointed to existing tables; Acknowledgment: Partially addressed for 5QgN, acknowledged by JvbR/CFSi).
* **Scalability:** The method adds a noticeable parameter count (~20%), potentially increasing inference costs compared to methods like LoRA (though less than full fine-tuning). (Initial concern: CFSi; Rebuttal: Not directly addressed w.r.t inference cost; Acknowledgment: Acknowledged by CFSi).
* **Marginal/Inconsistent Improvements:** Gains over baselines, particularly on downstream tasks, were sometimes seen as marginal or inconsistent, although the method generally showed the best performance preservation. (Initial concern: JvbR, Vwee; Rebuttal: Focused on trade-off; Acknowledgment: Acknowledged by JvbR/Vwee).

In light of the above, the AC recommends acceptance.